# Higher Risk of Acute Respiratory Distress Syndrome and Risk Factors among Patients with COVID-19: A Systematic Review, Meta-Analysis and Meta-Regression

**DOI:** 10.3390/ijerph192215125

**Published:** 2022-11-16

**Authors:** Yi-Tseng Tsai, Han-Chang Ku, Sujeewa Dilhani Maithreepala, Yi-Jing Tsai, Li-Fan Chen, Nai-Ying Ko, Sriyani Padmalatha Konara Mudiyanselage

**Affiliations:** 1Department of Nursing, An-Nan Hospital, China Medical University, Tainan 709, Taiwan; 2Department of Nursing, College of Medicine, National Cheng Kung University, 1 University Road, Tainan 70101, Taiwan; 3Department of Nursing, Faculty of Allied Health Sciences, University of Peradeniya, Peradeniya 20400, Sri Lanka; 4Department of Nursing, National Cheng Kung University Hospital, College of Medicine, Tainan 704, Taiwan; 5Operation Theatre Department, The National Hospital of Sri Lanka, Colombo 00700, Sri Lanka

**Keywords:** acute respiratory distress syndrome, coronavirus infection, risk factor

## Abstract

Objective: To estimate the global risk and risk factors associated with acute respiratory distress syndrome (ARDS) among patients with COVID-19: Design: A systematic review, meta-analysis and meta-regression. Setting and Participants: Hospitals or nursing homes and patients with acute respiratory distress syndrome after COVID-19. Methods: The literature review was systematically conducted on Embase, MEDLINE, CINAHL, and Web of Science, in addition to manual searches and reference list checking from 1 January 2019 to 2 March 2022. The search terms included coronavirus, acute respiratory syndrome, acute respiratory distress syndrome and observational studies. Three reviewers independently appraised the quality of the studies and extracted the relevant data using the Joanna Briggs Institute abstraction form and critical appraisal tools. A study protocol was registered in PROSPERO (CRD42022311957). Eligible studies were meta-analyzed and underwent meta-regression. Results: A total of 12 studies were included, with 148,080 participants. The risk ratio (RR) of ARDS was 23%. Risk factors were age ≥ 41–64 years old (RR = 15.3%, 95% CI =0.14−2.92, *p* = 0.03); fever (RR = 10.3%, 95% CI = 0.03−2.03, *p* = 0.04); multilobe involvement of the chest (RR = 33.5%, 95% CI = 0.35–6.36, *p* = 0.02); lymphopenia (RR = 25.9%, 95% CI = 1.11–4.08, *p* = 0.01); mechanical ventilation with oxygen therapy (RR = 31.7%, 95% CI = 1.10–5.25, *p* = 0.002); European region (RR = 16.3%, 95% CI = 0.09–3.17, *p* = 0.03); sample size ≤ 500 (RR = 18.0%, 95% CI = 0.70–2.89, *p* = 0.001). Conclusions and Implications: One in four patients experienced ARDS after having COVID-19. The age group 41–64 years old and the European region were high-risk groups. These findings can be used by policymakers to allocate resources for respiratory care facilities and can also provide scientific evidence in the design of protocols to manage COVID-19 worldwide.

## 1. Introduction

The novel coronavirus disease 2019 (COVID-19) is associated with severe acute respiratory syndrome coronavirus 2 (SARS-CoV-2) and severe acute respiratory syndrome (SARS), which caused the global pandemic of COVID-19 in late 2019 and which is continuing to date [1,2,3]. The rapid increase in COVID-19 has critically influenced society, healthcare systems, and people worldwide [4,5]. The clinical spectrum of disease presentation could be asymptomatic, mild, moderate, or severe, with some cases leading to death. In addition, some patients develop severe lung failure (acute respiratory distress syndrome, ARDS) [3,6,7]. Previous research data showed that 76·40% of cases in Greece [8] and 61.7% in Germany [9] have developed ARDS in high proportions. However, data also show lower rates of 3.60% in Poland [10], and the admission rate to intensive care units (I.C.U.) is 11.5%, as high as in [11]. It is also evident that patients with moderate-to-severe ARDS need invasive mechanical ventilation and a wide range of therapeutic actions [10,12].

Furthermore, patients who recovered after ARDS may have a decline in exercise capacity and health-related quality of life [10]. Previous literature showed that 30% of hospital admissions among COVID-19 patients experienced ARDS [13], and some among them had experienced the insertion of extracorporeal membrane oxygenation (ECMO) after COVID-19 [14]. They understand the global risk of ARDS among patients with COVID-19 to allocate resources and other healthcare management of post-recovery ARDS patients with COVID-19. Furthermore, some research studies found that age, sex, and previous illness history, such as hypertension, diabetes mellitus, chronic obstructive pulmonary disease, asthma, and chronic kidney failure, may influence the incidence of ARDS among patients with COVID-19 [15,16,17,18]. However, no studies have systematically included all risk factors. Moreover, no systematic review and meta-analysis studies were conducted using the primary research data to describe the risk of ARDS and risk factors among patients having COVID-19 from a global perspective. Therefore, this study aims to identify the global risk and the risk factors associated with acute respiratory distress syndrome among patients with COVID-19.

### 1.1. The Objective of This Study 

This study aimed to estimate the global risk and risk factors associated with acute respiratory distress syndrome among patients with coronavirus infection.

### 1.2. Research Question

What are the global risk and risk factors associated with acute respiratory distress syndrome among patients with coronavirus infection?

## 2. Methods

### 2.1. Search Strategy 

This study protocol was registered with the International Prospective Register of Systematic Reviews (PROSPERO Reg No—CRD42022311957). Five databases (Embase, MEDLINE, CINAHL, and Web of Science) were searched for studies on the prevalence of acute respiratory distress syndrome among patients with coronavirus published between 1 January 2019 and 2 March 2022. The Preferred Reporting Items for Systematic Reviews and Meta-Analyses guidelines (PRISMA) were followed [19,20,21,22]. English synonyms including coronavirus, acute respiratory disease, 2019-nCoV, COVID-19, SARS-CoV, respiratory syndrome, severe acute respiratory syndrome, severe acute respiratory infection, Middle East respiratory syndrome, hospital, hospitalization, hospitalized, inpatient, patient, and sufferer were used to search each database. Several control vocabularies for the Emtree and MeSH databases were also used. For Emtree, these included “coronavirus”, “Wuhan seafood market pneumonia virus”, “Wuhan coronavirus”, “Hospital patient”, “Adult respiratory distress syndrome”, “severe acute respiratory syndrome”, “ARDS”, and “SARS”. We supplemented the search results with the Endnote X9 bibliographical database. Publications that cited the papers identified during the search and the reference lists of relevant articles and previous systematic reviews were manually screened to confirm the sensitivity of the search strategy. 

### 2.2. Inclusion and Exclusion Criteria

The inclusion criteria were: (1) the study provided primary data on the prevalence of acute respiratory distress syndrome (ARDS) measured using validated assessment tools or coded medical report data within a population-based study after COVID-19 occurred; (2) participants were diagnosed with COVID-19; and (3) the studies were observational, such as cohort and cross-sectional studies, and were published in English, Chinese, or Sinhala from 2019 to 2022. The following types of studies were excluded: studies for which the study population did not include COVID-19 patients and studies that were qualitative research and review articles. 

Titles and abstracts were independently screened by three researchers based on the inclusion and exclusion criteria after removing duplicates using the Endnote X9 bibliographical database. Then, the full texts of the selected studies were reviewed by three researchers independently, with any disagreement resolved by a fourth researcher to avoid selection bias.

### 2.3. Quality Assessment

All eligible studies were assessed for quality of evidence using the Joanna Briggs Institute (J.B.I.) Critical Appraisal for Checklist for Prevalence Studies Scale (CACPSS), which contains nine items and four responses (yes, no, unclear, and not applicable) [23]. Studies with a total score of 8 and above were considered acceptable quality evidence and were included in this systematic review and meta-analysis. Study quality and risk of bias were also independently assessed by the three reviewers, with any disagreement resolved by a fourth researcher. 

### 2.4. Data Extraction

The following data were extracted: names of authors, year of publication, country, settings, study design, sample size, participant ethnicity, age, gender, and prevalence of acute respiratory distress syndrome. In addition, three authors independently assigned quality scores for the included studies according to the PRISMA guidelines [20,21,22], and any disagreements were resolved via a discussion among all four authors.

### 2.5. Statistical Analysis 

A meta-analysis was conducted to identify statistical outcomes of higher risk of ARDS among patients with COVID-19 using the eligible studies. The pooled prevalence of acute respiratory distress syndrome was analyzed using several events converted to the risk ratio of 95% CI and *p*-values and a fitted model based on heterogeneity. Random or fixed-effects models were used based on the heterogeneity of results for acute respiratory distress syndrome among coronavirus infection patients. We transformed the proportions with the Freeman–Tukey double arcsine method before pooling the data for the prevalence rate of acute respiratory distress syndrome [24]. The heterogeneity value was assessed using I^2^, Cochran’s Q test, and Tau2 for the included studies according to the DerSimonian–Laird estimator [25,26,27,28]. A zero value indicated the absence of heterogeneity, 25% indicated no significance or low significance, 50% indicated moderate heterogeneity, and 75% indicated significant heterogeneity. In the present study, 75–100% indicated significant heterogeneity, where the Q statistic and *p*-value were used to validate the heterogeneity results. In this meta-analysis, I^2^ < 75% and *p* < 0.05 indicated statistical significance.

Publication bias was determined using funnel plots, and the Q statistic for Egger’s test was used to determine the correlation between the effect estimate and the variances in the results for acute respiratory distress syndrome via C.M.A. software and a visual examination of the funnel plots [29,30]. A subgroup analysis and a meta-regression were performed to investigate potential sources of heterogeneity. For the meta-regression, we used the pool of effect size data as a single covariable introduced individually into the models. A simultaneous test was conducted to determine if all coefficients were zero in the model test. We used a null hypothesis model for the effect size comparison. Statistical analyses were conducted using Comprehensive Meta-Analysis Software version 3.0 (Biostat, Englewood, NJ, USA.) [23].

## 3. Results

### 3.1. Study Identification

A total of 35,005 articles published from 1 January 2019 to 2 March 2022 were identified during the initial database search. Six thousand seven hundred eighty-seven articles were removed as duplicates using the Endnote X9 bibliographical database. The titles and abstracts of 28,220 articles were screened, and 1611 articles met the inclusion criteria.

The full text of each article was read to determine eligibility, and 1596 articles were excluded due to the following reasons: 469 articles did not have any relationship to COVID; 1099 articles did not mention ARDS behaviour among COVID-19 patients; 25 articles did not assess the outcome variables; and 3 articles were not available in a full-text format. In addition, three articles were removed after the quality assessment due to a low-quality score in the peer review. Finally, 12 articles were included in the systematic review and meta-regression (Figure 1). Studies with quality scores of 8 and above were accepted as high quality (Appendix A).

### 3.2. Study Characteristics

Characteristics of the 12 included studies are shown in Table 1. The included studies were conducted in 7 countries. Four were conducted in China [31,32,33,34], two were in the United States [15,35] and Germany [3,9], and only one was from Greece [8], India [36], Poland [10], and Korea [37]. Regarding the study design used, six were retrospective studies [10,15,31,33,34,35], three were cross-sectional studies [3,8,9], two were cohort studies [32,37], and one was a prospective study [36]. Regarding the study setting, 12 studies were conducted in a hospital. These studies were published between 2020 and 2022 (Table 1). 

### 3.3. Participant Characteristics 

The total participants in the 12 studies were 148,080 individuals; 74,3851 were male, and 72,860 were female. The participant age range in 12 studies was 30–70 years of age, and one study [36] did not mention the participant’s age.

In terms of ethnicity, five studies were conducted using Chinese populations [32,33,34,36,38], five studies were conducted in Caucasian populations [3,8,9,10,35], and one study from the United States also mentioned five ethnic groups (African American, Caucasian, American Indian, Asian, Mixed) [15]. One study was conducted in a Korean population [37]. Studies were conducted in 4 WHO regions: five studies were conducted in the Western Pacific region [31,32,33,34,37], four studies were conducted in the European region [3,8,9,10], two studies were conducted in the region of the Americas [15,35], and one study was conducted in the South-East Asian region [36] (Table 1).

### 3.4. Higher Risk of ARDS among Patients with COVID-19 

Within the seven countries (the United States, Germany, Greece, India, China, Poland, and Korea), 12 studies analyzed the higher risk of acute respiratory distress syndrome among patients with COVID-19. The reported numbers were 7320 of 140760 participants, and four studies in the European region showed the highest rates of ARDS (Table 1 and Table 2). After conducting a meta-analysis, we found that the pooled risk ratio of ARDS among patients with COVID-19 was 23% (95% CI = 14.3–34.7%, *p* = 0.001), with significant heterogeneity within the 12 studies (I² = 99.70, Q = 3685.601, Tau2 = 1.002, *p* = 0.001, Figure 2).

### 3.5. Risk Factors of ARDS among COVID-19 through Meta-Regression Analysis

Based on the meta-analysis results, we identified significant heterogeneity within outcome variables of risk of ARDS. Therefore, a meta-regression analysis was conducted to identify factors affecting heterogeneity through the subgroups. The meta-regression model included the following risk factors for ARDS among patients with COVID-19: gender, age, smoking, cluster exposure history, fever, muscular soreness, cough, productive cough, sore throat, dyspnea, fatigue, headache, diarrhea, nausea and vomiting, lung infiltrates or consolidation, multilobe involvement in the chest, leucopenia, lymphopenia, underlying illnesses, diabetes mellitus, hypertension, chronic obstructive pulmonary disease, asthma, chronic kidney failure, chronic cardiac disease, coronary artery disease, thyroid disease, antiviral therapy, antibiotic therapy, nasal cannula oxygen therapy, mechanical ventilation oxygen therapy, WHO region, and sample size. The results regarding acute respiratory distress syndrome among coronavirus infection patients for statistical model 1, random effects, Z-distribution, and the log odds ratio. The model test was a simultaneous test to confirm that all coefficients (excluding the intercept) were zero (Q = 3685.601, df = 12, *p* = 0.00). 

The following risk factors were identified as having a significant relationship with ARDS for patients with coronavirus infection: age greater than or equal to 41–64 years old (RR = 15.3%, 95% CI =0.14−2.92, *p* = 0.03); fever (RR = 10.3%, 95% CI = 0.03−2.03, *p* = 0.04); multilobe involvement in the chest (RR = 33.5%, 95% CI = 0.35–6.36, *p* = 0.02); lymphopenia (RR = 25.9%, 95% CI = 1.11–4.08, *p* = 0.01); mechanical ventilation with oxygen therapy (RR = 31.7%, 95% CI = 1.10–5.25, *p* = 0.002); European region (RR = 16.3%, 95% CI = 0.09–3.17, *p* = 0.03); and sample size less than or equal to 500 (RR = 18.0%, 95% CI = 0.70–2.89, *p* = 0.001) (Table 2).

### 3.6. Publication Bias 

Publication bias was analyzed using a funnel plot and Egger’s test on ARDS among patients with COVID-19 for the 12 studies. However, the funnel plot did not show evidence of asymmetry, and there was a minor probability of publication bias. Statistically, possible publication bias was observed based on Egger’s test results (Q = 3685.6, *p* = 0.001, I^2^ = 99.70%, Figure 3) due to the diversity of the sample sizes and the length of the publication time in the included studies.

## 4. Discussion

This systematic review and meta-analysis evaluated the higher risk of ARDS among patients with COVID-19 during the recent pandemic of COVID-19. A comprehensive search for relevant studies yielded 12 studies from 7 countries across the four WHO regions (region of the Americas, the Southeast Asian region, the European region, and the Western Pacific region) from 2020 to 2022. Additionally, this study has identified risk factors for ARDS among patients with COVID-19 to provide scientific evidence for respective stakeholders to prepare and allocate resources for any future pandemics.

According to our findings, 23% of COVID-19 patients experienced ARDS. This means that nearly one in four patients had progressed to ARDS. Those who had COVID-19 needed to have advanced care plans for further treatments because severe respiratory failure with the progress of ARDS is a possible complication of COVID-19 infection. Additionally, it may relate to the cause of death in COVID-19. Therefore, well-organized and reliable observational systems are beneficial for the patient screening process to detect ARDS early among patients with COVID-19, especially in people in residential care facilities. This will aid in early transfer to specialized medical care units for the proper management of ARDS to minimize the risk of death and severe complications, such as lung fibrosis or permanent lung damage. Most importantly, the early identification of high-risk patients will be the better choice for timely, evidence-based treatments and approaches to prevent further post-COVID-19 complications [9,15,35,37]. 

Furthermore, this study analyzed possible risk factors associated with ARDS in COVID-19 patients. Age greater than or equal to 41–64 years was a significant risk factor for ARDS among patients with COVID-19. Our results were consistent with previous studies, with similar significant factors associated with the poor prognosis of COVID-19 [34,38,39,40]. In addition, as per previous literature, medical co-mobilities were associated with the risk of ARDS in middle-aged adults who had COVID-19, specifically pre-existing respiratory disease. However, we did not find any significant association between medical co-morbidities and ARDS [3,40]. Therefore, a pre-preparedness lifestyle modification strategy needs to be applied for future prevention plans for COVID-19 for middle-aged adults around the globe. An example of this would be effective communication for information distribution during pandemics within adult communities to mitigate their myths about infectious diseases such as COVID-19 in the future [41,42]. 

According to our analysis, fever, multilobe involvement in the chest, lymphopenia, and mechanical ventilation with oxygen therapy were significant clinical risk factors for ARDS among patients with COVID-19. This is because most respiratory distress patients need multifaceted ventilation systems and frequent position changes, such as prone positioning and vital sign monitoring. Therefore, clinicians and healthcare people need to arrange facilities for a long-time inpatient care management strategy for their clinical units [9,15,31,32]. It should be more suitable for nursing home care facilities to arrange respiratory care resources for a future convention. 

### 4.1. Strengths

This study has several strengths. First, we have included studies from four WHO regions with different populations and ethnicities, such as black, white, Chinese, and Indian. This was the first systematic review and meta-regression for ARDS among patients with COVID-19 to include risk factors. Therefore, we recommend future studies should include the Eastern Mediterranean region with Arabic and Islamic populations and the African region with the black African community to see if there is any difference in risk of ARDS and risk factors. This is because, in our findings, the European region is one of the significant risk factors for ARDS among patients with COVID-19 [3,8,9]. In our study, when to start mechanical ventilation and how mechanical ventilation processes proceeded during the clinical management were also found to be risk factors for ARDS. Therefore, it is essential to identify the oxygenation peak flow measurement during ventilation periods, such as from the starting point until the end. Additionally, continuous blood saturation monitoring and advanced technological methods for blood oxygenation, such as extracorporeal membrane oxygenation (ECMO), are recommended [3,14,43,44,45].

### 4.2. Limitations

There were some limitations to our study. First, we noticed a possible publication bias due to the diversity of the sample sizes and the length of the publication time in the included studies. Therefore, we need to include a large sample with long-term observational studies, such as prospective cohort studies, throughout the pandemic. At this time, the available published studies were limited to 12 within seven countries due to early publication. Future analysis can be focused more on studies with new treatment strategies, such as ECMO, for ARDS and their survival rate. Most of the patients in the included studies are still in the hospital under treatment. Therefore, post-COVID co-morbidity for ARDS among patients with COVID-19 need to be analyzed in the future rehabilitation of patients with ARDS. 

## 5. Conclusions 

One in four patients has a risk of ARDS after acquiring COVID-19. The risk factors included middle-aged adults older than or equal to 41–64 years old, fever, multilobe involvement in the chest, lymphopenia, and mechanical ventilation with oxygen therapy. Additionally, the European region is at a high risk of ARDs among COVID-19 patients. Therefore, this study’s findings are beneficial for frontline clinicians, healthcare clinical decision-makers, and health policymakers to precisely justify the healthcare system and government of COVID-19 to arrange early interventions and suitable treatment strategies. 

This study provides scientific evidence to support clinical practice and the design of protocols to prevent ARDS. It is also a reference for future researchers who plan to examine ARDS and the risk factors among diverse populations. We recommend that future studies focus on the Eastern Mediterranean and African regions with multiple co-morbidities. 

## Figures and Tables

**Figure 1 ijerph-19-15125-f001:**
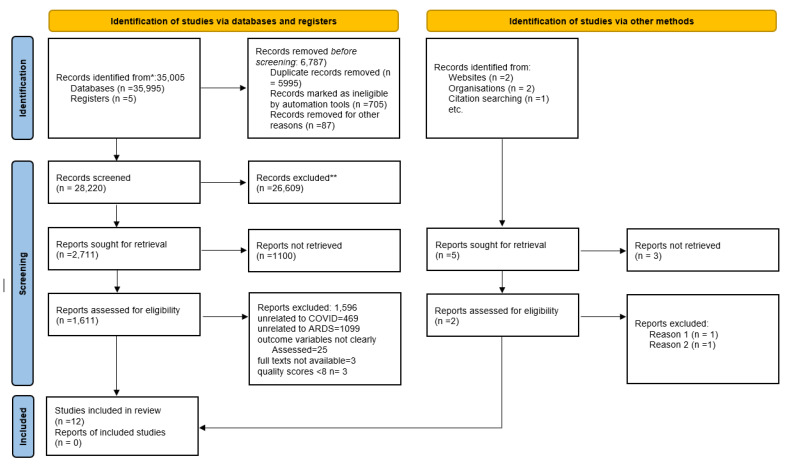
PRISMA flow diagram of the included studies.

**Figure 2 ijerph-19-15125-f002:**
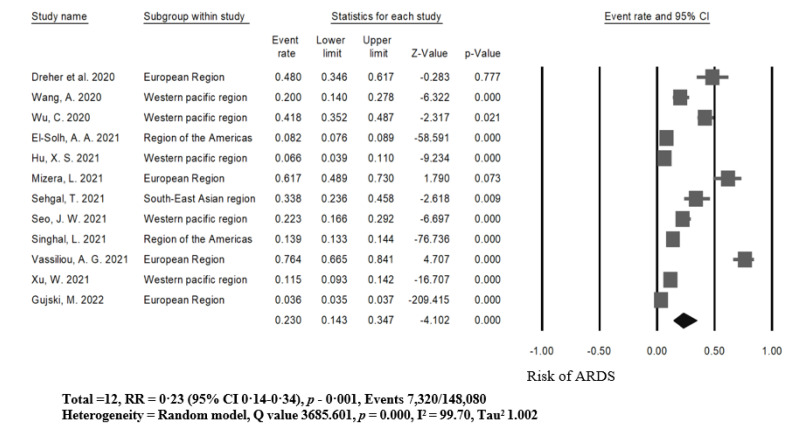
Prevalence rate of acute respiratory distress syndrome (ARDS) in patients with COVID-19 [3,8,9,10,15,31,32,33,34,35,36,37].

**Figure 3 ijerph-19-15125-f003:**
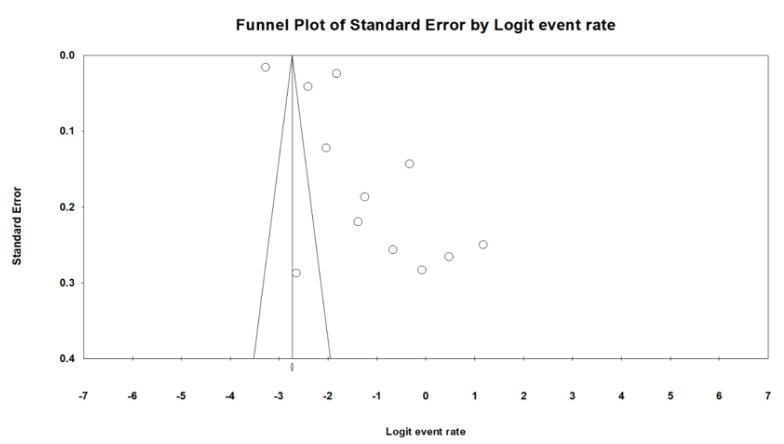
Funnel plots describing publication bias based on ARDS in patients with COVID-19.

**Table 1 ijerph-19-15125-t001:** Characteristics of the included studies.

Author, (Year) Country	WHO Region	Study Design	Ethnicity (n)	Mean Age	Sample Size (n)	Gender (n)	Prevalence Rate of ARDS (n) (%)	Joanna Briggs Institute Score
Dreher et al. (2020) Germany [3]	European region	Cross- sectional study	Caucasian: 50	ALL: 65. ARDS: 62. No ARDS: 68.	50	Male—33 Female—17	24 (48%)	9
Wang et al. (2020) China [31]	Western Pacific region	Retrospective study	Chinese: 130	ARDS: 63.5. No ARDS: 40.	130	Male—17 Female—128	26 (20%)	9
Wu et al. (2020) China [32]	Western Pacific region	Cohort study	Chinese: 201	ARDS: 58.5. No ARDS: 48.	201	Male—128 Female—73	84 (41.80%)	8
El-Solh et al. (2021) United States [35]	Region of the Americas	Retrospective study	Caucasian:3547 Black: 3264 Latino:989	ARDS: 69 No ARDS: 70.	7816	Male—7387 Female—60	643 (8.23%)	9
Hu et al. (2021) China [34]	Western Pacific region	Retrospective study	Chinese: 197	ALL: 45. ARDS: 58. No ARDS: 42.	197	Male—93 Female—104	13 (6.60%)	9
Mizera et al. (2021) Germany [9]	European region	Cross- sectional study	Caucasian: 60	ARDS: 69.1. No ARDS: 63.3	60	Male—36 Female—24	37 (61.67%)	8
Sehgal et al. (2021) India [36]	South-East Asian region	Prospective study	Chinese: 68	NA	68	Male—43 Female—25	23 (33.82%)	9
Seo et al. (2021) Korea [37]	Western Pacific region	Cohort study	Korean: 166	ARDS: 72. No ARDS: 56	166	Male—78 Female—88	37 (22.29%)	8
Singhal et al. (2021) United States [15]	Region of the Americas	Retrospective study	African: 4089 Caucasian:7462 Indian: 258 Asian: 427 Mixed race: 4 Unknown: 2544	ARDS: 62. No ARDS: 68.	14,785	Male—7358 Female—7427	2052 (13.88%)	8
Vassiliou et al. (2021) Greece [8]	European Region	Cross- sectional study	Caucasian: 89	ARDS: 62. No ARDS: 61.	89	Male—70 Female—19	68 (76.40%)	9
Xu et al. (2021) China [33]	Western Pacific region	Retrospective study	Chinese: 659	ARDS: 56.5. No ARDS: 49.	659	Male—332 Female—327	76 (11.53%)	9
Gujski et al. (2022) Poland [10]	European Region	Retrospective study	Caucasian: 116,539	NA	116,539	Male—60,915 Female—55,624	4237 (3.60%)	9

**Table 2 ijerph-19-15125-t002:** Meta-analysis according to subgroup and meta-regression used to identify factors affecting heterogeneity within the selected studies.

	Meta-Analysis					Meta-Regression			
Variable	No of Study	Sample	Risk Ratio (%) (95% CI)	*p-*Value	I²	Coefficient	Standardized Error	95% CI	*p-*Value
Gender	12	147711	23.8 (17.7–31.2)	0.001	99.42				
Female	12	72860	22.8 (14.5–33.9)	0.001	99.05	Reference			
Male	12	74851	25.1 (14.7–39.4)	0.001	99.60	0.1048	0.4521	−0.78–0.99	0.8166
Age	5	138642	10.0 (6.2–15.7)	0.001	99.66				
14–40 years old	5	3,642	12.4 (2.6–42.3)	0.001	99.61	Reference			
≥41–64 years old	5	51207	13.5 (6.7–25.4)	0.001	99.61	1.5309	0.7094	0.14–2.92	**0.0309**
≥65 years old	3	49793	4.4 (2.6–7.3)	0.001	95.73	1.1377	0.8132	−0.45–2.73	0.1618
Smoking	4	9498	14.1 (9.4–20.5)	0.001	95.37				
No smoking	4	5587	15.5 (6.8–31.5)	0.001	97.49	Reference			
Smoking	4	3911	12.5 (5.5–25.7)	0.008	79.48	−0.2463	0.7439	−1.70–1.21	0.7406
Cluster exposure history	2	735	15.9 (8.7–27.4)	0.001	85.31				
No exposure history	2	277	16.8 (12.8–21.7)	0.290	10.68	Reference			
Exposure history	2	458	20.5 (2.9–68.7)	0.001	91.78	0.2372	0.8990	−1.52–1.99	0.7919
Fever	7	10139	14.8 (9.9–21.7)	0.001	96.16				
No fever	7	4404	8.5 (4.7–15.1)	0.001	75.26	Reference			
Fever	7	5735	20.9 (12.0–33.7)	0.001	97.09	1.0355	0.5113	0.03–2.03	**0.0429**
Muscular soreness	3	1022	11.5 (6.8–18.8)	0.001	83.12				
No muscular soreness	3	842	13.8 (6.6–26.5)	0.001	91.80	Reference			
Muscular soreness	3	180	8.7 (5.3–14.0)	0.438	0.00	−0.5387	0.6182	−1.75–0.67	0.3835
Cough	7	10029	22.2 (15.2–31.3)	0.001	96.51				
No cough	7	7010	18.0 (9.2–32.1)	0.001	93.50	Reference			
Cough	7	3019	26.8 (15.6–42.1)	0.001	96.50	0.5185	0.5164	−0.49–1.53	0.3153
Productive cough	5	1317	19.3 (12.5–28.5)	0.001	91.20				
No productive cough	5	775	16.6 (8.7–29.6)	0.001	91.86	Reference			
Productive cough	5	542	22.1 (11.3–38.7)	0.001	91.72	0.3575	0.5552	−0.73–1.44	0.5196
Sore throat	4	1090	14.2 (8.5–22.8)	0.001	85.01				
No sore throat	4	957	17.3 (8.9–31.0)	0.001	92.93	Reference			
Sore throat	4	133	10.0 (5.9–16.7)	0.654	0.00	−0.7391	0.6484	−2.01–0.53	0.2543
Dyspnea	4	9044	19.9 (11.2–32.9)	0.001	97.86				
No dyspnea	4	6097	8.6 (2.9–22.8)	0.001	95.91	Reference			
Dyspnea	4	2947	38.5 (11.4–75.4)	0.001	98.04	1.9171	1.0015	−0.45–3.88	0.0556
Fatigue	3	8381	9.1 (7.2–11.4)	0.001	76.55				
No fatigue	3	6787	10.1 (9.4–10.9)	0.182	41.29	Reference			
Fatigue	3	1594	8.6 (5.2–13.9)	0.010	78.34	−0.1430	0.2636	−0.65–0.37	0.5876
Headache	3	1029	20.3 (11.0–34.5)	0.001	89.51				
No headache	3	876	19.6 (10.0–34.9)	0.001	91.03	Reference			
Headache	3	153	23.0 (3.5–71.4)	0.001	92.12	0.1005	0.8026	−1.47–1.67	0.9003
Diarrhea	6	9854	14.7 (10.5–20.0)	0.001	90.32				
No diarrhea	6	8936	14.9 (9.2–23.2)	0.001	94.83	Reference			
Diarrhea	6	918	11.1 (9.1–13.5)	0.018	63.50	−0.0371	0.4417	−0.90–0.82	0.9330
Nausea and vomiting	3	1022	13.6 (8.7–20.5)	0.001	77.02				
No nausea and vomiting	3	912	12.2 (6.8–20.9)	0.001	87.59	Reference			
Nausea and vomiting	3	110	16.3 (6.6–35.0)	0.088	58.81	0.3524	0.5670	−0.75–1.46	0.5342
Lung infiltrates or consolidation	3	460	14.7 (4.9–36.9)	0.001	90.59				
No lung infiltrates or consolidation	3	225	6.9 (4.1–11.1)	0.966	0.00	Reference			
Lung infiltrates or consolidation occurred	3	235	24.7 (6.0–62.9)	0.001	91.70	1.4722	1.1172	−0.71–3.66	0.1876
Multilobe involvement in chest	2	366	7.3 (1.6–28.1)	0.001	90.70				
Without multilobe involvement in chest	2	116	0.8 (0.1–5.8)	0.986	0.00	Reference			
With multilobe involvement in chest	2	250	19.7 (4.3–57.2)	0.001	95.14	3.3557	1.5330	0.35–6.36	**0.0286**
Leucopenia	2	327	14.9 (7.1–28.6)	0.003	78.69				
Without leucopenia	2	83	18.1 (11.2–27.9)	0.714	0.00	Reference			
With leucopenia	2	244	12.4 (3.1–39.0)	0.001	91.60	−0.5511	1.0741	−2.65–1.55	0.6079
Lymphopenia	2	327	11.1 (2.6–37.4)	0.001	93.32				
Without lymphopenia	2	227	3.1 (1.5–6.4)	0.614	0.00	Reference			
With lymphopenia	2	100	30.4 (12.2–5.8)	0.011	84.47	2.5987	0.7571	1.11–4.08	0.0006
Underlying medical illnesses	4	1103	32.7 (15.8–55.8)	0.001	95.48				
Without underlying medical illnesses	4	593	34.6 (10.5–70.6)	0.001	95.34	Reference			
With underlying medical illnesses	4	510	31.2 (9.3–66.6)	0.001	95.99	–0.1561	1.0793	−2.27–1.95	0.8850
Diabetes mellitus	10	131086	22.7 (16.3–30.6)	0.001	98.69				
Without diabetes mellitus	10	119438	17.8 (10.0–29.4)	0.001	98.97	Reference			
With diabetes mellitus	10	11648	29.9 (18.9–43.8)	0.001	96.92	0.7022	0.4680	–0.21–1.61	0.1335
Hypertension	10	131100	20.6 (14.9–27.7)	0.001	98.68				
Without hypertension	10	111673	15.2 (8.5–25.9)	0.001	98.34	Reference			
With hypertension	10	19427	28.5 (16.5–44.5)	0.001	98.59	0.7917	0.4925	−0.17–1.75	0.1079
Chronic obstructive pulmonary disease	8	130591	20.8 (14.1–29.7)	0.001	98.53				
Without chronic obstructive pulmonary disease	8	125717	19.4 (11.0–32.0)	0.001	99.15	Reference			
With chronic obstructive pulmonary disease	8	4874	21.4 (12.0–35.3)	0.001	89.10	0.3264	0.5407	−0.73–1.38	0.5461
Asthma	3	777	25.4 (12.6–44.4)	0.001	84.17				
Without asthma	3	766	19.4 (8.8–37.6)	0.001	90.94	Reference			
With asthma	3	11	49.1 (18.7–80.2)	0.317	12.85	1.3206	0.9398	−0.52–3.16	0.1600
Chronic kidney failure	8	131013	20.0 (13.8–28.3)	0.001	98.48				
Without chronic kidney failure	8	125421	19.9 (11.7–31.9)	0.001	99.16	Reference			
With chronic kidney failure	8	5592	19.1 (10.6–31.8)	0.739	91.53	0.1008	0.5133	−0.90–1.10	0.8443
Chronic cardiac disease	7	130816	13.1 (9.3–18.2)	0.001	98.86				
Without chronic cardiac disease	7	99673	13.0 (6.5–24.3)	0.001	99.39	Reference			
With chronic cardiac disease	7	31143	11.2 (7.1–17.2)	0.048	92.89	0.1858	0.5377	−0.86–1.23	0.7296
Coronary artery disease	3	258	43.6 (25.4–63.7)	0.001	83.69				
Without coronary artery disease	3	235	34.5 (16.9–57.8)	0.001	90.22	Reference			
With coronary artery disease	3	23	62.4 (35.3–83.4)	0.262	25.28	1.1040	0.8250	−0.51–2.72	0.1808
Thyroid disease	2	247	25.5 (19.5–32.2)	0.159	42.11				
Without thyroid disease	2	236	26.0 (16.1–39.2)	0.073	68.96	Reference			
With thyroid disease	2	11	32.0 (8.5–70.5)	0.179	44.66	0.2440	0.9489	−1.61–2.10	0.7971
Antiviral therapy	2	398	24.0 (7.4–55.3)	0.001	94.74				
Without antiviral therapy	2	331	35.6 (4.9–85.6)	0.001	93.64	Reference			
With antiviral therapy	2	67	15.4 (1.6–66.6)	0.001	97.38	−1.1104	1.7311	−4.50–2.28	0.5212
Antibiotic therapy	2	495	15.0 (3.5–46.3)	0.001	94.37				
Without antibiotic therapy	2	144	7.8 (2.9–19.0)	0.255	22.77	Reference			
With antibiotic therapy	2	351	20.3 (2.8–68.9)	0.001	95.14	0.8311	1.5353	−2.17–3.84	0.5883
Nasal cannula oxygen therapy	2	398	20.1 (4.6–56.7)	0.001	96.64				
Without nasal cannula oxygen therapy	2	139	42.9 (10.7–82.5)	0.001	94.20	Reference			
With nasal cannula oxygen therapy	2	259	7.8 (1.3–34.9)	0.001	92.14	−2.1811	1.3269	−4.78–0.41	0.1002
Mechanical ventilation oxygen therapy	4	15272	47.0 (19.1–76.9)	0.001	99.69				
Without mechanical ventilation oxygen therapy	4	13425	17.4 (5.3–44.3)	0.001	98.63	Reference			
With mechanical ventilation oxygen therapy	4	1847	83.9 (48.3–96.7)	0.001	87.16	3.1785	1.0584	1.10–5.25	0.0027
WHO region	12	148080	23.0 (14.3–34.7)	0.001	99.70				
Region of theAmericas	2	25296	10.7 (6.3–17.6)	0.001	99.34	Reference			
European region	4	121104	39.3 (4.2–90.5)	0.001	99.53	1.6336	0.7856	0.09–3.17	0.0376
Western Pacific region	5	1589	18.0 (9.1–32.4)	0.001	96.10	0.6010	0.7562	−0.88–2.08	0.4268
Southeast Asianregion	1	91	33.8 (23.6–45.8)	1.00	0.00	1.4473	1.1285	−0.76–3.65	0.1997
Sample size	12	148080	23.0 (14.3–34.7)	0.001	99.70				
>500	4	146807	8.4 (3.7–18.1)	0.088	99.89	Reference			
≤500	8	1273	35.7 (21.3–53.2)	0.001	95.33	1.8006	0.5585	0.70–2.89	0.0013

CI, confidence interval.

## Data Availability

Data sharing does not apply to this article as no datasets were generated or analyzed during the current study.

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
