# Peer review of "Higher Risk of Acute Respiratory Distress Syndrome and Risk Factors among Patients with COVID-19: A Systematic Review, Meta-Analysis and Meta-Regression"

_ijerph, 2022, doi:10.3390/ijerph192215125_

Round 1
Reviewer 1 Report
I consider that the study is relevant and well presented. However, the statistical analysis description, from my point of view, is vague. More details about the Freeman-Tukey double arcsine method could help to understand the way in which the study was conducted. The same suggestion applies for I2 , Cochran Q test, Tau2 and DerSimonian-Laird estimator, mentioned by the authors to measure the heterogeneity. Additionally, a justification about why this methods and concepts were chosen among others would be relevant for researchers interested in follow similar studies.
Author Response
Reviewer 1
I consider that the study is relevant and well presented. However, the statistical analysis description, from my point of view, is vague. More details about the Freeman-Tukey double arcsine method could help to understand the way in which the study was conducted. The same suggestion applies for I2 , Cochran Q test, Tau2 and DerSimonian-Laird estimator, mentioned by the authors to measure the heterogeneity. Additionally, a justification about why this methods and concepts were chosen among others would be relevant for researchers interested in follow similar studies.
Response to Reviewer 1
Thank you for the comments and we agreed with you. we change and edit our reference again please refer to the reference section

Reviewer 2 Report
In this manuscript, authors have collected and reviewed recent publications then analyzed the risk and risk factor of ARDS in COVID-19 patients globally. As the results there are some indicators that may help clinical personnel to manage and prevent possible ARDS development in similar outbreaks.
Overall, the design of this study is sound and data been collected are in fair quality. However, as I checked one of the 12 studies authors mentioned (El-Solh, et al), the race is not all “white”. There are three populations indicated as Caucasians, Blacks, and Latinos. Which brings up the possibility that authors’ conclusion about “white population” or “European origin” may be incorrect. Authors need to re-define the sub populations in the study carefully.
The writing style and the presentation of results need to be unified. There are a lot of misspellings and typos all over the manuscript. For example, in line 66, it is patients, not patents; in line 130, p ”<” 0.05, not ”>” 0.05.
In the reference section, there are many duplicates such as #2 and #37; #35 and #44; #42 and #43, these are critical errors for publications.
Other significant confusions are in Figure 1., the numbers in first two boxes are incorrect. In Table 2, all the descriptions at the top are not clear due to overlays.
Authors need to proof read their manuscript before submission.
Author Response
Reviewer 2
In this manuscript, authors have collected and reviewed recent publications then analyzed the risk and risk factor of ARDS in COVID-19 patients globally. As the results there are some indicators that may help clinical personnel to manage and prevent possible ARDS development in similar outbreaks.
Overall, the design of this study is sound and data been collected are in fair quality. However, as I checked one of the 12 study’s authors mentioned (El-Solh, et al), the race is not all “white”. There are three populations indicated as Caucasians, Blacks, and Latinos. Which brings up the possibility that authors’ conclusion about “white population” or “European origin” may be incorrect. Authors need to re-define the sub populations in the study carefully.
Response to reviewer 2
Thank you for the comments. we change total number of Caucasian:3547, Black: 3264, Latino:989 in table one. We check out all the data again; however, still European region is the high risk but not the ethnicity of the white population.
The writing style and the presentation of results need to be unified. There are a lot of misspellings and typos all over the manuscript. For example, in line 66, it is patients, not patents; in line 130, p ”<” 0.05, not ”>” 0.05.
Response to reviewer 2
Thank you for the comments. We change all misspellings by using the English editing system. Please refer to line 66 and line 130
In the reference section, there are many duplicates such as #2 and #37; #35 and #44; #42 and #43, these are critical errors for publications.
Response to reviewer 2
Thank you for the comments. We updated all references again, and please refer reference section
Other significant confusions are in Figure 1., the numbers in first two boxes are incorrect. In Table 2, all the descriptions at the top are not clear due to overlays.
Authors need to proof read their manuscript before submission.
Response to reviewer 2
Thank you for the comments. We updated all table and figure again and please table and figure
Note: this manuscript sends to the National Cheng Kung University English proofreading centre

Reviewer 3 Report
Thank you authors for the submission, a few comments with the current work.
1. Abstract: methodology mentioned 2 reviewer, in the full text, it was mentioned 3 reviewer, please clarify.
2. English and grammar: require moderate edit especially within the introduction and discussion section.
3. Several spelling mistakes, and some editing that was left to be deleted, please address that.
4. Table 2: too crowded, some label are not readable. Please re-do, with relevant sectioning.
Objective, methodology and discussions points were otherwise good. Good luck with your submission.
Author Response
Reviewer 3
- Abstract: methodology mentioned 2 reviewer, in the full text, it was mentioned 3 reviewer, please clarify.
Response to reviewer 3
Thank you for the comments. We change as three authors in the abstract. Please refer abstract in line 22
- English and grammar: require moderate edit especially within the introduction and discussion section.
Response to reviewer 3
Thank you for the comments. this manuscript has been sent to the National Cheng Kung University English proofreading centre this time
- Several spelling mistakes and some editing that was left to be deleted, please address that.
Response to reviewer 3
Thank you for the comments. this manuscript has been sent to the National Cheng Kung University English proofreading centre this time
- Table 2: too crowded, some label are not readable. Please re-do, with relevant sectioning.
Response to reviewer 3
Thank you for the comments. All the figures and tables are organized and edited
Response to reviewer 3
Thank you for the comments
Objective, methodology and discussion points were otherwise good. Good luck with your submission.
Response to reviewer 3
Thank you for the comments

Round 2
Reviewer 2 Report
Thanks to the authors for their corrections and updates. However, the formation, spelling, and punctuation marks in the revised version still need to be unified to improve the reading experiences. For example: 1) in the abstract there are multiple unnecessary colons in line 16, 17, 18, and 25; 2) in line 45, the number should be 76.40%; 3) in line 266 there is half a word (oxygenation) been crossed over. 4) in line 285, what is "study fining"? 5) There are missing information in reference number 2 (line 313). Moreover, due to the correction of duplicated references, the citation number in the main article doesn't match the reference section (such as the citations in line 168). Needless to say, the numbers mentioned in the manuscript are often not able to add up. Especially in line 165, figure 1, and table 1. Such kind of mistakes and inconsistencies appears everywhere in the manuscript, I strongly suggest the authors carefully proof read the whole article, includes tables and figures, thoroughly before next re-submission.
By the way, authors included ref. 37 as a "Chinese population", but the study was performed in Korea. This kind of ethnic group definition error should be carefully avoid.
Overall, the quality of this manuscript needs more improvement to fulfill requirements of this journal.
Author Response
Response to Reviewer 2
Thanks to the authors for their corrections and updates. However, the formation, spelling, and punctuation marks in the revised version still need to be unified to improve the reading experiences. For example: 1) in the abstract there are multiple unnecessary colons in line 16, 17, 18, and 25; 2) in line 45, the number should be 76.40%; 3) in line 266 there is half a word (oxygenation) been crossed over. 4) in line 285, what is "study fining"? 5) There are missing information in reference number 2 (line 313).
Response to reviewer 2
Thank you for the comments. We changed all the formation, spelling, and punctuation marks. Please refer to lines 16,17, 18, and 25. Also, line 45.
We change oxygenation and finding. Please refer lines 266 and 285
Moreover, due to the correction of duplicated references, the citation number in the main article doesn't match the reference section (such as the citations in line 168). Needless to say, the numbers mentioned in the manuscript are often not able to add up. Especially in line 165, figure 1, and table 1. Such kind of mistakes and inconsistencies appears everywhere in the manuscript, I strongly suggest the authors carefully proof read the whole article, includes tables and figures, thoroughly before next re-submission.
Response to reviewer 2
Thank you for the comments. We changed and edited it again as per your comments. Please refer to lines 168 and 165 and citations, tables and figures.
We do proofreading again
By the way, authors included ref. 37 as a "Chinese population", but the study was performed in Korea. This kind of ethnic group definition error should be carefully avoid.
Response to reviewer 2
Thank you for the comments. We changed and edited it again as per your comments
Overall, the quality of this manuscript needs more improvement to fulfil requirements of this journal.
Response to reviewer 2
Thank you for the comments. We changed and edited it again as per your comments.
